# Papillomaviruses and Papillomaviral Disease in Dogs and Cats: A Comprehensive Review

**DOI:** 10.3390/pathogens13121057

**Published:** 2024-12-01

**Authors:** John S. Munday, Cameron G. Knight

**Affiliations:** 1School of Veterinary Science, Massey University, Palmerston North 4442, New Zealand; 2Faculty of Veterinary Medicine, University of Calgary, Calgary, AB T3R 1J3, Canada; cgknight@ucalgary.ca

**Keywords:** papillomavirus, review, dog, cat, cancer, carcinogenesis, oncogenesis, feline papillomavirus, canine papillomavirus, CPV, FcaPV, viral carcinogenesis, warts, viral plaques, Bowenoid in situ carcinoma, squamous cell carcinoma

## Abstract

Papillomaviruses (PVs) frequently infect humans as well as non-human species. While most PV infections are asymptomatic, PVs can also cause hyperplastic papillomas (warts) as well as pre-neoplastic and neoplastic lesions. In this review, the life cycle of PVs is discussed, along with the mechanisms by which PVs cause hyperplastic and neoplastic diseases. The humoral and cell-mediated immune responses to PVs are reviewed, giving context to the later discussion on the use of vaccines to reduce canine and feline PV-associated disease. Both dogs and cats are infected by numerous different PV types classified into multiple different PV genera. The taxonomic classification of PVs is reviewed, along with the significance of this classification. The PV-associated diseases of dogs and cats are then described. These descriptions include the clinical presentation of the disease, the causative PV types, the histological features that allow diagnosis, and, where appropriate, possible treatment options. The review is comprehensive and contains the latest information about PVs and the diseases they cause in dogs and cats.

## 1. Introduction

In addition to causing human hyperplastic papillomas (warts), papillomaviruses (PVs) are also estimated to cause around 4% of all cancers in people [1]. While these cancers are caused by only a small subset of the over 200 different human PV types, the sexually-transmitted high-risk alpha PVs cause most cervical squamous cell carcinomas (SCCs) as well as a significant proportion of human penile and oral SCCs. In dogs, PVs cause viral papillomas (warts), but in contrast to humans, PVs appear to be a rare cause of canine neoplasia. Viral papillomas are extremely rare in cats, but there is increasing evidence that PVs may be an under-recognized cause of neoplasia in this species. In this review, the latest literature regarding PVs is discussed, along with a description of the lesions associated with PV infection in dogs and cats. 

## 2. Papillomaviral Life Cycle and Carcinogenesis

Papillomaviruses (PVs) are double-stranded circular DNA viruses. Their genome contains between five and seven early (E) genes and two or, rarely, three late (L) genes [2]. For the overwhelming majority of PV types, viral replication is coordinated with the division and maturation of cells within stratified squamous epithelium [3,4]. Infection occurs when microtrauma allows a viral particle to contact the basal cells of the epithelium [5]. The PV gains entry and expression of *E1* and *E2*, allowing small numbers of viral copies to be produced. As the basal cells replicate, the PV DNA is also replicated within the basal cell layer, allowing the persistence of infection [6,7]. However, infectious viral particles are only produced when an infected basal cell terminally differentiates [4,8]. When this happens, *E6* and *E7* are expressed, and the resultant proteins stimulate continued replication of both the infected epithelial cells and PV DNA [9]. As the infected cell reaches the superficial layers of the epithelium, the L1 and L2 capsid proteins are expressed, and as the epithelial cells slough, assembled viral particles are released into the environment [3].

Viral replication is only possible because the PV proteins can induce the normally post-mitotic suprabasilar epithelial cells to continue to divide. This ability of PVs to influence cell growth is illustrated by the development of viral papillomas (warts). In these lesions, PV infection results in rapid epithelial cell division, causing marked thickening of the epithelium, subsequent folding, and the development of an exophytic wart. This ability of PVs to promote cell growth and division is also the basis of their ability to influence the development of neoplasia. 

The mechanisms by which the human ‘high-risk’ PVs cause neoplasia have been well-studied in humans [9]. The PV E6 and E7 proteins are most important, and *E6* and *E7* are often referred to as the PV oncogenes. However, it should be remembered that although most human PV types produce E6 and E7 proteins, only a small proportion of types cause cancer [10]. In humans, the key factor in determining PV oncogenesis by the high-risk PV types is the accidental integration of PV DNA into the cell DNA [4]. Integration is considered a random occurrence, and most infections result in only variable degrees of epithelial hyperplasia. In these infections, the viral DNA remains episomal, and the production of PV E6 and E7 proteins is well-regulated to maximize viral replication. However, the integration of viral DNA into the cell DNA can result in the unregulated production of PV oncoproteins [4]. The E6 protein of human high-risk PVs prevents cell apoptosis by degrading the p53 tumor suppressor protein, activating telomerase proteins, allowing replication of older cells, as well as altering several pathways important in cell signaling [11]. The E7 protein promotes cell division by degrading proteins important in cell cycle control, especially retinoblastoma protein (pRb), allowing greater cell replication [11]. Loss of pRb also results in a marked increase in cell p16^CDKN2A^ protein (p16) that is so consistent that p16 immunostaining is used as a marker of PV-induced neoplasia [12]. In addition, the E7 protein predisposes to DNA replication errors by disrupting the centrosome cycle [13]. Therefore, unregulated expression of high-risk type PV *E6* and *E7* creates a rapidly dividing population of older epithelial cells that are resistant to apoptosis and are prone to inaccurate DNA replication. This creates ideal conditions for the additional cell DNA mutations required for neoplastic transformation. While some low-risk PV types have been associated with human skin cancers, the PV DNA is not typically integrated, and these neoplasms do not contain increased p16 immunostaining [14,15]. 

While the PV type is the most important factor in determining the oncogenic risk of a human PV infection, host factors are also important. For example, there is individual variability in the time taken to mount an immune response against PV infection. A rapid response will quickly clear a PV infection, making it less likely that the PV DNA will be accidentally integrated into the cell DNA. In contrast, an immunosuppressed person will have a more prolonged PV infection, a higher number of infected cells, and a subsequent increased chance of accidental integration [16]. Environmental cofactors can also be important in determining the result of PV infection. While cofactors are less important for cancers caused by the high-risk types in humans, skin cancers caused by low-risk human PV types are highly dependent on sun exposure as a cofactor [10]. This is because the low-risk PVs prevent apoptosis but do not cause significant DNA mutation as these types do not cause DNA mutation; they do not, by themselves, cause cancer. However, exposure of an infected cell to sunlight causes DNA mutations, and, as apoptosis cannot be triggered, the mutations accumulate, resulting in cell transformation. 

The mechanisms by which PVs induce neoplasia in dogs and cats are less well understood [17]. However, studies of canine viral plaques and feline Merkel cell tumors show that, as in humans, integration of viral DNA is likely important in progression to neoplasia [18,19]. Similarly to human cancers, the promotion of cell replication due to E7-induced degradation of pRb has been demonstrated in PV-induced cancers of dogs and cats [20]. As in humans, the loss of pRb causes a marked increase in p16, which can be detected using immunohistochemistry (Figure 1) [20,21]. In contrast to the established role of E7 in PV oncogenesis in dogs and cats, the role of the E6 protein is less clear. While Canis familiaris papillomavirus (CPV) type 2 caused warts that progressed to neoplasia in a group of immunodeficient dogs [22], the E6 protein of this PV does not degrade p53 [23]. In cats, in vitro evidence suggests the Felis catus papillomavirus (FcaPV) type 2 E6 protein may degrade p53 [24]. Additionally, loss of p53 was reported in a series of feline Merkel cell carcinomas that contained FcaPV2 DNA [25]. However, loss of p53 immunostaining was not associated with the presence of FcaPV2 DNA in a series of feline cutaneous SCCs [20]. Unlike the high-risk human PVs, there is no evidence that either canine or feline PVs influence telomerase activity [26]. Host factors are important in some types of PV-associated neoplasms, with some breeds of dogs and cats having much higher rates of cancer development [16,27,28]. Furthermore, exposure to environmental factors, especially UV light, is likely to be important for some PV-associated skin cancers, especially in cats [29]. 

## 3. Immune Response to Papillomavirus Infection

There are two components of the immune response to PV infection. The humoral response results in the production of antibodies against the PV [30]. These antibodies effectively prevent further infection by this PV type, and this protection is the basis for using vaccines to prevent infection by the PV types that cause cancer [31]. The presence of antibodies does not clear an existing PV infection, and dogs have been reported to have both warts and very high antibody titers against the causative PV type [32]. 

To resolve an existing PV infection, the body has to generate a cell-mediated immune response to detect and destroy infected cells [10]. The generation of a cell-mediated response is the cause of the spontaneous resolution that is seen with PV-induced papillomas [33]. As the time taken to the onset of the cell-mediated immune response is variable [10], the time of viral papilloma resolution also shows individual variability [34]. Differences between hosts in their ability to prevent viral replication and resolve infection are probably important in determining why some infected animals, but not others, develop PV-induced neoplasia [35,36]. Interestingly, some canine PV types have also been shown to be able to reduce the immune response against them [37].

## 4. Classification of Papillomavirus Types

With the notable exception of the ruminant Deltapapillomaviruses, PVs are highly species-specific, and PVs are named according to their host species. PVs are classified into different genera (<60% similarity) and types (<90% similarity) using the highly conserved *L1 ORF* [38]. Both dogs and cats are infected by numerous PV types classified within multiple genera (Figure 2). The classification system is clinically useful as PV types within the same genus often cause similar lesions (Table 1) [38]. 

Currently, the *Papillomavirus episteme* (https://pave.niaid.nih.gov/#home; accessed 22 October 2024) contains 24 Canis familiaris papillomavirus (CPV) types and 7 Felis catus papillomavirus (FcaPV) types. In addition, other types have been partially sequenced from both species [53], suggesting additional types will be fully sequenced in the future. None of the canine or feline PV types are known to have any zoonotic potential, and there is no conclusive evidence that human PV types can infect dogs or cats. 

## 5. Consequences and Diagnosis of Papillomavirus Infection

Most PV infections in dogs and cats are asymptomatic, with studies showing that a high proportion of both dogs and cats are infected by PVs without showing any clinical signs [54,55]. While additional research is required, it appears animals are likely to be asymptomatically infected by some PV types at an early age and remain infected throughout life. Rarely, changes in host factors may allow increased viral replication, and previously asymptomatic infections can become clinically visible as pre-neoplastic or even neoplastic lesions (Figure 3). 

Initial infection by some Lambdapapillomavirus or Taupapillomavirus types can result in rapid viral replication and the development of a viral papilloma (wart) [56]. The development of an immune response prevents rapid replication and causes lesion regression, although whether the PV is completely cleared from the body or persists within basal cells is not currently clear [34]. 

Cross-species infection of cats by Bos taurus PV (BPV) type 14 (a Deltapapillomavirus) can result in a neoplasm of mesenchymal cells referred to as a feline sarcoid [57]. There is no evidence that BPV14 can persist asymptomatically in feline skin [58], although the proportion of cats that develop sarcoids after infection by BPV14 is unknown. Although putative BPV DNA was recently detected in some canine samples [59], dogs are generally not considered susceptible to Deltapapillomavirus infection, and there is currently little evidence that PVs cause canine mesenchymal neoplasia.

Hyperplastic lesions caused by PVs can generally be diagnosed on gross examination when multiple exophytic lesions develop on the skin or in the mouth of a younger dog [41]. Confirmation of a PV etiology in these cases can performed using histology to detect the PV-induced cell changes as described in the following description of viral papillomas. Immunohistochemistry using antibodies against the PV L1 protein can also be used [32]. However, care must be taken as cross-reactivity between PV types cannot be assumed. Additionally, L1 proteins are only produced when viral replication occurs, which may not always be present, especially in neoplastic disease [4]. As described, immunohistochemistry to detect increased cellular p16 protein is routinely used in human pathology to indicate a PV etiology of some cancers, and this technique also appears to indicate a PV etiology of some feline and canine neoplasm types [12,21,42]. Papillomaviral DNA can also be amplified from lesions using PCR. However, due to the frequent presence of asymptomatic PV infection [14,54,55], the detection of PV DNA or RNA within a lesion does not prove that PV caused the lesion. Likewise, while in situ hybridization techniques can demonstrate the presence of PV DNA or RNA within a lesion [31], definitively proving causality is difficult, especially for neoplasms that do not show PV-induced cell changes or increased p16 protein. 

## 6. Papillomaviral Lesions in Dogs

### 6.1. Canine Viral Papillomas

Viral papillomas (warts) are very common in dogs, with many cases suspected to develop without the owner noticing or the dog being presented to a veterinarian [60]. As these develop at the time of first infection by the PV, most of these are seen in younger dogs, typically between 6 months and two years of age. Increased exposure to infectious PV particles also appears important, and there are rare reports of outbreaks of viral papillomas, for example, in doggie daycare facilities [61]. Due to the role of microtrauma in PV entry into the epithelium, viral papillomas are most common in the mouth, on the feet, and around the ears and face. They typically cause little discomfort to the dog, although rarely, they can cause lameness or difficulty eating [62]. Viral papillomas are most often caused by CPV1, 2, or 6, although other CPV types and mixed infections have been reported within these lesions [41]. Papillomaviruses have also been reported to cause corneal and nailbed viral papillomas in dogs [63,64]. 

Clinical examination reveals proliferative, exophytic, often multiple masses (Figure 4). Careful examination often reveals a roughened surface or finger-like projections that develop due to the folding of the epithelium. Canine cutaneous viral papillomas can rarely be covered by a thick layer of grey keratin (cutaneous horn) [65]. Lesions can be subdivided into exophytic and endophytic (inverted) papillomas. These subtypes are due to different folding patterns of the hyperplastic epithelium with an endophytic papilloma appearing as thickened epithelium protruding from a sunken cup-like mass [66]. Whether a viral papilloma is exophytic or endophytic does not appear to be determined by the causative PV type and does not influence the expected clinical behavior [67]. Histology reveals marked thickening of the epithelium, typically with PV-induced changes within the cells (Figure 5). These changes include enlarged cells with increased quantities of blue-grey granular cytoplasm, cells with nuclei that are dark, shrunken, and surrounded by a clear halo (koilocytes), and intranuclear eosinophilic viral inclusions. Le Net papillomas are a subtype of canine cutaneous warts that contain florid histological evidence of PV infection [68]. These do not show different clinical behavior, and while additional cases are required, in one of the author’s experiences (JSM), these are likely to be due to CPV6 infection. 

Viral papillomas develop as a normal response by normal epithelial cells to PV proteins, and although these lesions have sometimes previously been referred to as neoplasms, they are more correctly considered hyperplastic [56]. Once the body generates a cell-mediated immune response, the wart resolves. The time taken for resolution is variable, and most papillomas resolve within 3–6 months, although warts can be present for up to two years before spontaneous resolution [34,69]. 

As canine viral papillomas are expected to resolve spontaneously and rarely cause discomfort, most are not treated. However, if treatment is required, it is necessary to destroy all infected cells by, for example, cryotherapy or surgical excision [70]. Removal of the wart is expected to be curative, although recurrence or development of additional warts can occur in the absence of an immune response. No medical therapy has been proven to hasten the spontaneous resolution of warts in dogs (or, indeed, in people). Warts on dogs treated with the antibiotic azithromycin resolved significantly faster than warts on untreated dogs in one study [71]. However, the apparent ‘benefit’ of treatment was not because warts on treated dogs resolved unexpectedly rapidly but because the untreated dogs had abnormally long-lasting warts [56]. Based on this single study, azithromycin has been widely used to treat canine papillomas. However, no subsequent reports have shown azithromycin to be beneficial for inducing or hastening wart regression in dogs [62,70,72]. Azithromycin is not used to treat warts in people. Therapeutic vaccination has been proposed as beneficial, both using autologous vaccines as well as virus-like particle capsid vaccines [73]. While such vaccines can stimulate high antibody titers, they have not been shown to be beneficial in stimulating the necessary cell-mediated immune response required for lesion resolution [32,70]. In a large study of people with genital warts, a more rapid resolution was not seen in response to vaccination [74]. Imiquimod cream is used to treat warts in people. While this treatment is likely to work for smaller cutaneous viral papillomas in dogs, it should be remembered that this medication works by inducing lymphocyte-mediated necrosis of epithelial cells [75]. The cream has no specific action against PVs, and so has no advantage over other techniques (such as cryotherapy or surgery) that destroy infected epithelial cells. Other therapeutics, including natural and homeopathic therapies, have been suggested as beneficial [76], but conclusive evidence of efficacy is hard to prove when treating lesions that spontaneously resolve. 

Very rarely do dogs develop persistent warts. While there is no clear definition of the clinical syndrome of ‘persistent warts’, the continued development of new warts six months after warts are initially observed and the progression to numerous large warts over an extensive area are key features (Figure 6) [32,77]. As previously noted, some cases of warts in dogs can remain small but be present for up to two years prior to spontaneous resolution. Warts, in such cases, do not meet the criteria for persistence. Persistent warts develop in dogs that are unable to mount an immune response against the PV infection. While this could, in theory, develop due to any immunosuppressive disease, dogs with persistent warts most often show no other signs of immunosuppression, suggesting an immune defect specific to PV infection. While the genetic causes of this have not been investigated in dogs, people with persistent warts (referred to in humans as recalcitrant warts or ‘tree man syndrome’) are thought to be unable to produce or stimulate CD4+ (T-helper) lymphocytes [78]. Unfortunately, the presumed canine immune defect means warts recur after surgical excision in affected dogs. In the authors’ experience, dogs with persistent warts are usually euthanatized due to the local effects of numerous large warts or, less frequently, due to progression to SCC [46,49]. However, a dog with extensive papillomas that had persisted for 22 months was recently reported to have been cured by a combination of surgical excision and radiation therapy [72].

### 6.2. Canine Viral Cutaneous Plaques

Viral cutaneous plaques (also referred to as pigmented plaques) are rare pre-neoplastic lesions of dogs that are caused by a number of closely related Chipapillomavirus types [79]. While additional research is required, evidence from humans and cats suggests it is likely that dogs are frequently asymptomatically infected by Chipapillomaviruses, and the development of lesions is primarily dependent on host factors allowing increased PV replication and subsequent lesion development. This mechanism has some similarities to the human condition of epidermodysplasia verruciformis, in which people develop cutaneous viral plaques due to a defect in the inherent keratinocyte immune reaction against PV infection [16]. A role for a genetically-mediated immune dysfunction is supported by rare reports of plaques developing in dogs with immunosuppressive conditions and the predisposition for plaque development observed in some breeds, especially Pugs and Vizslas [16,80,81,82,83]. The likely genetic predisposition suggests breeding from affected animals should be discouraged, although plaques usually develop later in life when animals are less likely to be used for breeding purposes. It appears likely that plaques do not develop when a dog is first infected, which means that, unlike papillomas, plaques tend to develop in middle-aged dogs [16,84]. 

Clinically, viral plaques initially appear as multiple, dark, 1–10 mm diameter, slightly raised plaques that are most common on the ventrum and medial aspects of the limbs (Figure 7) [83,85]. Rarely do plaques progress to become exophytic lesions present over a large proportion of the body [84]. Plaques confined to the pinna have also been reported [86]. Histology reveals a moderately thickened epidermis covered by increased keratin. Lesions typically have a scalloped appearance with prominent epidermal and dermal pigmentation and keratohyalin granules (Figure 8). In contrast to feline viral plaques, PV-induced cellular changes are infrequently present in canine plaques [79]. Canine cutaneous plaques occasionally form keratin-filled cysts that can be mistaken for follicular tumors [62]. 

Most canine viral plaques remain small and only of cosmetic concern. However, extensive plaques can cause pruritus or pain [84]. Progression to SCC is generally considered a rare event, although such progression was reported in almost 10% of cases in a study of viral plaques that had been submitted for histological diagnosis [83]. Plaques caused by CPV16 may be most likely to progress to neoplasia [44,87]. Evidence from human PV-induced skin cancers suggests exposure to the sun could be a factor in the progression from canine viral plaque to SCC. 

Treatment of dogs with pigmented plaques is often not required. However, as with papillomas, treatment is aimed at destroying the infected epithelial cells, for example, by surgical excision, cryotherapy, laser therapy, or tigilanol tiglate gel [81,88]. As plaque development is likely caused by an inability of the host to control viral infection, investigation for potential underlying immunosuppressive conditions may be appropriate. Medical treatments such as oral retinol, topical imiquimod, azithromycin, and interferon alfa-2b have been proposed, but none are likely to restore normal immune function, and none have been conclusively shown to have any therapeutic benefit. 

### 6.3. Papillomavirus-Associated Neoplasia in Dogs

Evidence that PVs can cause SCCs in dogs is derived from the rare progression of cutaneous viral plaques or persistent oral papillomas to SCCs [44,77,89]. Additionally, CPV17 DNA was detected in an oral SCC that showed histological evidence of PV-induced cell changes and intense p16 immunostaining, supporting a viral cause (Figure 9) [42,90]. While studies of both oral and cutaneous canine SCCs have demonstrated that a proportion contain PV DNA [91,92,93], the frequent asymptomatic infection by PVs makes the significance of this uncertain [54]. Currently, there is no evidence that PVs are a common factor in the development of either oral or cutaneous SCCs in dogs [94,95]. 

In addition to SCCs, PVs appeared to be likely factors in the development of multiple follicular neoplasms in a dog [96]. Evidence of a PV etiology included the consistent amplification of CPV3 (a Chipapillomavirus) DNA from the neoplasms, as well as the concurrent presence of viral pigmented plaques containing CPV3 DNA. Studies of other types of canine cancer, including oral melanoma, nailbed SCC, urothelial carcinoma, pulmonary adenocarcinoma, and mammary tumor, did not reveal the presence of PV DNA [97,98,99]. 

## 7. Papillomaviral Lesions in Cats

### 7.1. Feline Viral Papillomas

Domestic cats rarely develop viral papillomas (warts), with only a handful of cases reported. These included oral viral papillomas that were detected as incidental findings under the tongue of cats and cutaneous viral papillomas on the nasal planum or eyelids (Figure 10) [46,47,100,101]. Both oral and cutaneous viral papillomas are most likely caused by FcaPV1 (a Lambdapapillomavirus), although a cutaneous papilloma was reported to contain DNA sequences that are likely from a novel, uncharacterized FcaPV type [102].

### 7.2. Feline Viral Cutaneous Plaques

Depending on the degree of dysplasia in the lesion, feline cutaneous viral plaques have been subdivided into viral plaques (early lesions with mild dysplasia) and Bowenoid in situ carcinomas (BISCs; later lesions with more marked dysplasia). Due to uncertainties regarding the degree of dysplasia required for differentiation and the absence of any known differences in biological behavior between the two lesions, this distinction is considered redundant, and both lesions are now simply considered viral plaques. As in dogs, feline viral plaques are pre-neoplastic lesions that have the potential to progress to SCC. Feline cutaneous plaques are most often caused by FcaPV2 (a Dyothetapapillomavirus), although the Taupapillomavirus FcaPV types can also cause these lesions [50,51,103]. Evidence suggests almost all cats are infected by FcaPV2 around the time of birth [55]. These infections are asymptomatic in most cats, but a poorly understood acquired immune deficiency appears to allow more rapid viral replication and the development of visible lesions. The cause of the immune defect is almost always unknown; however, as Sphinx and Devon Rex cats develop plaques more frequently and at a younger age, the deficiency appears, at least in these breeds, to be genetically mediated [27,28]. 

Feline viral plaques present as multiple pigmented or non-pigmented sessile lesions up to 2 cm in diameter. Plaques are most common on the face, head, and neck, although they have been described in most areas of the body (Figure 11) [104]. As in dogs, as the plaques become bigger, they can become more exophytic, and some plaques may be covered in a hard crust composed of keratin and cell debris. Histology reveals mild to moderate epidermal hyperplasia with dysplasia of cells within the deeper layers of the epidermis. PV-induced cellular changes can be prominent in smaller plaques (Figure 12), although they are less frequently visible in larger, more dysplastic lesions [104,105]. Compared to canine viral plaques, feline viral plaques typically contain greater dysplasia, suggesting the PV types that cause these lesions in cats are more able to alter normal cell regulation than the canine Chipapillomavirus types. Feline viral plaques contain intense intranuclear and intracytoplasmic p16 immunostaining, supporting a PV etiology [20,21]. 

Compared to canine viral plaques, feline viral plaques are more likely to increase in size and number, resulting in significant morbidity [105]. Viral plaques in cats also appear much more likely to progress to SCC, especially in Devon Rex and Sphinx cats, in which the progression tends to be rapid and results in SCCs with high metastatic potential [27,28]. Whether or not sunlight plays a significant role in neoplastic transformation is uncertain. 

As in dogs, treatment of feline plaques is local rather than systemic, with surgical excision, laser ablation, imiquimod cream, and cryotherapy, which are all suggested therapies [88,105]. Considering the apparent role of altered host defenses in lesion development, additional lesions should be expected. 

### 7.3. Papillomavirus-Associated Neoplasia in Cats

Papillomaviruses were first associated with feline cutaneous SCCs in 2006 [106], and evidence of a causative association has been accumulating since this time. A feline cutaneous SCC should be considered to have been caused, at least partially, by PV infection when PV DNA is detectible within the SCC *and* the neoplastic cells exhibit intense nuclear and cytoplasmic p16 immunostaining (Figure 13). Using these criteria, PVs may cause as many as 75% of SCCs within UV-protected areas of the body (haired pigmented skin) and as many as 30% of SCCs from UV-exposed areas of the body (non-pigmented nasal planum, pinnae, and eyelids) [107]. Feline cutaneous SCCs can develop as a progression from a viral cutaneous plaque, although whether all PV-associated SCCs develop this way is currently uncertain [27,28]. 

As with feline viral cutaneous plaques, FcaPV2 is generally the PV type that is most often detected in feline cutaneous SCCs. However, both FcaPV3 and FcaPV4 are also thought to influence SCC development, and these may be the predominant type present in SCCs in some countries [107,108]. Histological evidence of PV infection is rarely detectible, and histology typically does not allow differentiation between SCCs that are caused by PV infection and those that are not. However, in addition to PCR amplification of viral DNA and p16 immunostaining, SCCs caused by PV infection also contain a higher copy number of PV DNA [49,109] and higher FcaPV gene expression within the neoplastic cells [109,110,111].

In addition to SCCs, PVs have also been associated with cutaneous basal cell carcinomas (BCC). This association was first detected when it was observed that the epidermis overlying a BCC often contained histological evidence of PV infection [85]. Further evaluation of BCCs revealed the presence of PV DNA, PV-induced cell changes within the neoplastic cells, and intense p16 immunostaining. Unlike SCCs, BCCs appear to be more frequently caused by non-FcaPV2 types, including FcaPV3 and FcaPV7, which have only been described in a BCC [50,53]. 

Studies of kittens and adult cats suggest most animals are infected by FcaPV2 in the first days of life, and infection with this PV type is lifelong [55]. While no similar studies have been performed on the other FcaPV types, asymptomatic infection is likely to be similarly common. Therefore, as with viral plaques, host factors are likely the most important factor in determining whether a PV infection will predispose to cancer. Interestingly, studies have revealed variability in the viral loads of FcaPV2 between cats. Furthermore, the viral loads on an individual cat remain constant over an extended period [112]. Whether cats with chronically high viral loads are predisposed to neoplasm development is unknown. Cancer development may also be dependent on the presence of co-factors, such as UV exposure [29,110]. 

In addition to epidermal neoplasia, FcaPV2 has also been reported to play a role in the development of feline cutaneous Merkel cell carcinomas [19,25,113]. Currently, only significant numbers of affected cats have been identified in Japan. Whether this is because of geographic differences in FcaPV2 or a genetic predisposition for this type of cancer in Japanese cats is uncertain. Alternatively, it is possible that different diagnostic criteria are used for skin neoplasia in different countries. Evidence supporting the role of FcaPV2 in the development of Merkel cell carcinoma includes the consistent detection of PV DNA and the presence of intense p16 immunostaining [25]. Interestingly, cats with Merkel cell carcinomas also often concurrently have viral plaques or PV-associated SCCs. 

The role of PVs in feline oral SCCs is currently unresolved. There is a single report of an oral in situ carcinoma containing FcaPV3 DNA, PV-induced cell changes, and intense p16 immunostaining [114]. However, larger studies of feline oral SCCs have produced conflicting results. Studies of feline oral SCCs in New Zealand and the United States detected PV DNA only rarely [115,116,117,118,119]. In contrast, FcaPV2 DNA was detected in 31% of oral SCCs from Italian cats, in around 15% of oral SCCs from Austrian cats, and in 43% of oral SCCs from Taiwanese and Japanese cats [120,121,122]. As with other cancers, it is difficult to exclude asymptomatic infection, and FcaPV2 DNA was also detected in the mouth of around 30% of cats without cancer in Italy. Unlike PV-associated SCCs of the skin, feline oral SCCs rarely contain intense nuclear and cytoplasmic p16 immunostaining, and although some variability in p16 immunostaining is present in these SCCs, this variability has not been associated with the presence of PV DNA (Figure 14) [117,123]. Overall, it appears PVs can infect the oral mucosa of cats; however, additional research is required to determine the significance of such infections in the development of feline oral SCCs. 

Feline sarcoids are also thought to be caused by PV infection in cats. In contrast to the other PV-associated feline neoplasms, sarcoids are mesenchymal neoplasms that are caused by cross-species infection by BPV14 [57]. These lesions are exophytic fleshy masses, most common around the face, especially around the nasal philtrum (Figure 15). A history of close contact with cattle is often reported, although sarcoids have been reported in cats without known contact with cattle [124,125]. Histology reveals a proliferation of mesenchymal cells underlying the hyperplastic epidermis, which extends into the dermis as thin rete pegs. Histological evidence of viral infection is not visible [126]. As BPV14 has not been detected as an asymptomatic infection of cats, the detection of this PV within a histologically consistent lesion allows a definitive diagnosis [58]. Sarcoids are locally infiltrative but have not been reported to metastasize. Treatment is complete excision of the neoplasm, but local recurrence can occur, especially when complete excision is not possible [124]. 

## 8. Vaccination to Prevent Papillomaviral Lesions in Dogs and Cats

As vaccines have been shown to prevent PV-induced disease in people [127], they may be similarly useful in the prevention of PV-induced disease in dogs and cats. Vaccines against both canine and feline vaccines have been produced experimentally, and the techniques used to produce human virus-like particle vaccines can be modified to produce vaccines against canine or feline PV types [31,112]. However, there are two important considerations for a vaccine to be a feasible prevention method. Firstly, the vaccine must be given prior to the first infection by the PV type [128]. Vaccination after infection has not been shown to reduce viral load or induce lesion resolution [74,112]. Secondly, for a vaccine to be commercially viable, the disease the vaccine is preventing must be common and result in significant morbidity or mortality. In humans, PV vaccines are primarily used to prevent infection by the high-risk sexually transmitted PV types. Vaccination against these is possible as people are not infected until they become sexually active, allowing a window in which to vaccinate prior to the first infection. Additionally, such vaccines are commercially viable due to the relatively frequent development and severe consequences of PV-induced cancer in people. 

Dogs are most likely first infected by the PVs that cause warts as young adults. This suggests it would be possible to vaccinate against these PVs prior to first exposure [31]. However, it is unlikely that owners would pay a significant amount of money to protect their dogs against a hyperplastic lesion that is self-resolving in almost all cases. Similarly, in humans, vaccines have not been developed to prevent cutaneous warts due to the low morbidity caused by these lesions. More research is required regarding when dogs are infected by the Chipapillomaviruses that cause viral plaques and, potentially, SCCs. Infection appears likely to occur early in life, reducing the opportunity for vaccination prior to infection. Furthermore, as viral plaques are rare in dogs, it is uncertain how many owners would pay to protect their dogs against this disease. 

Evidence suggests that cats are infected by the PV types that cause neoplasia in the first days of life, making vaccination prior to the first infection almost impossible [55]. Therefore, while a FcaPV2 vaccine was able to generate high antibody titres, it was not effective at reducing viral loads in vaccinated cats [112]. This suggests that such a vaccine would not influence whether that PV infection is likely to predispose to neoplasia.

## 9. Conclusions

Papillomaviruses are an important cause of non-neoplastic and neoplastic diseases in humans. While there are still some gaps in our knowledge regarding PVs in dogs and cats, research in the last 5 years has greatly expanded our knowledge of the diversity of PV types that infect these species as well as the variety of lesions caused by PV infections. 

## Figures and Tables

**Figure 1 pathogens-13-01057-f001:**
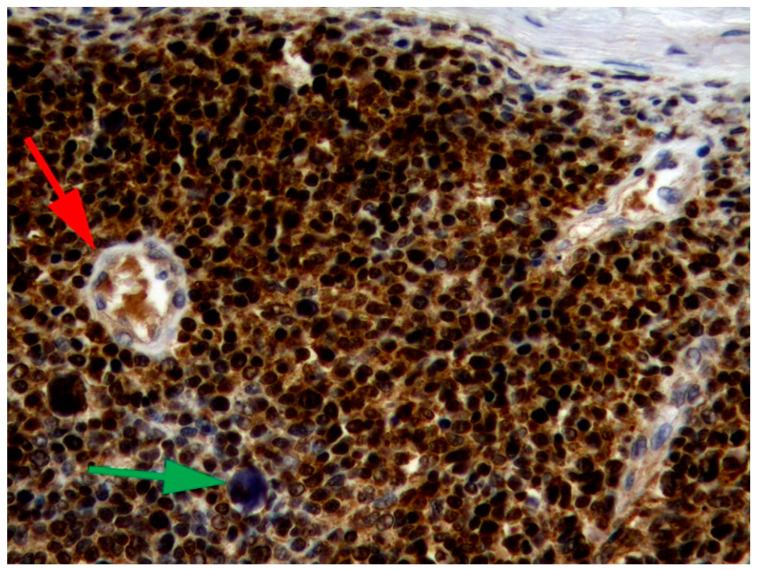
Papillomavirus-associated cutaneous basal cell carcinoma, cat. The neoplasm shows intense intracytoplasmic and intranuclear immunostaining using antibodies against the p16^CDKN2A^ protein. For comparison, immunostaining is not visible in smooth muscle or endothelial cells of blood vessels within the neoplasm (red arrow). This neoplasm contained marked papillomavirus-induced cell changes that are visible as basophilic smudged expanded cell cytoplasm (green arrow). Felis catus papillomavirus type 7 was amplified from the lesion. 400×.

**Figure 2 pathogens-13-01057-f002:**
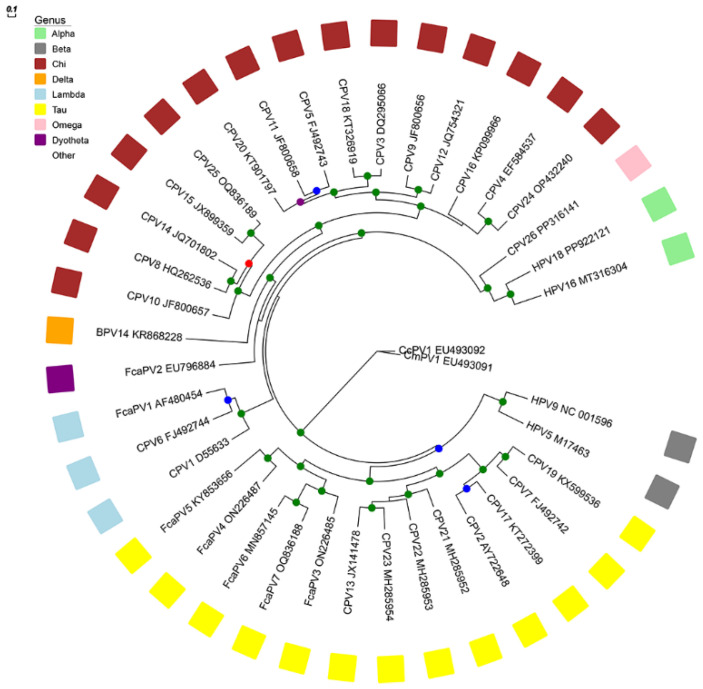
Phylogenetic tree. Unrooted Maximum likelihood phylogeny based on concatenated nucleotide alignment of E1, E2, L1 and L2 ORF sequences of the feline and canine papillomavirus (PV) types. Accession numbers for the sequences used are included. Abbreviations used include Canis familiaris, CPV; Bos taurus papillomavirus, BPV; Felis catus papillomavirus, FcaPV; human papillomavirus, HPV; Capreolus capreolus papillomavirus, CcPV, Chelonia mydas papillomavirus. CmPV. The PV genera are also listed. Internal branches are colored based on inferred branch support values, as determined by 1000 replicates usin RAxML. The scale bar indicates the genetic distance (nucleotide substitutions per site). Diagram courtesy of Dr. Matt Knox, Massey University.

**Figure 3 pathogens-13-01057-f003:**
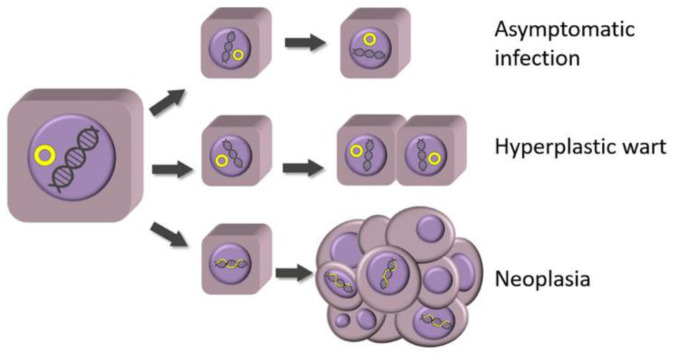
Schematic diagram illustrating possible outcomes of papillomavirus infection. Most papillomavirus infections result in minimal cell proliferation, and these infections remain asymptomatic. A small number of papillomavirus types induce marked proliferation of infected epithelial cells, resulting in a hyperplastic viral papilloma (wart). Accidental integration of the papillomaviral DNA into the host cell DNA can result in uncontrolled expression of viral proteins, predisposing to the development of a neoplastic lesion.

**Figure 4 pathogens-13-01057-f004:**
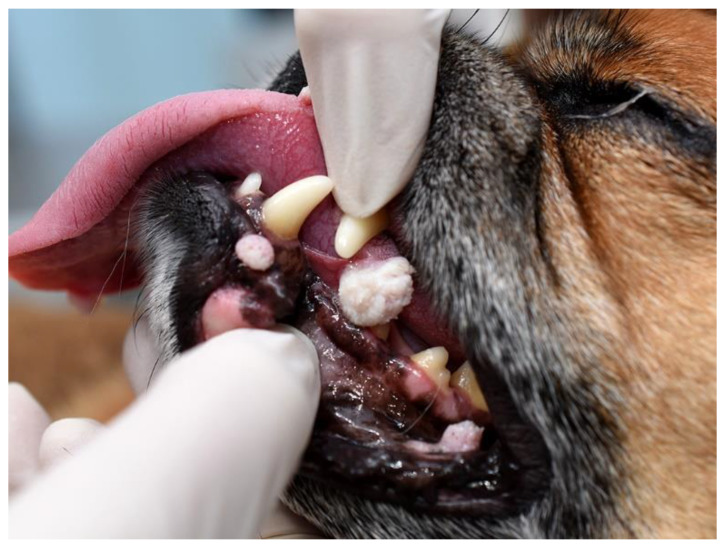
Canine oral warts. These warts are hyperplastic lesions caused by increased replication of epithelial cells due to papillomaviral proteins. As in this case, warts typically remain small and often have a roughened or feathered surface due to the thickened, folded epithelium. Photo courtesy Dr. Anne Quain, Sydney School of Veterinary Science, Australia.

**Figure 5 pathogens-13-01057-f005:**
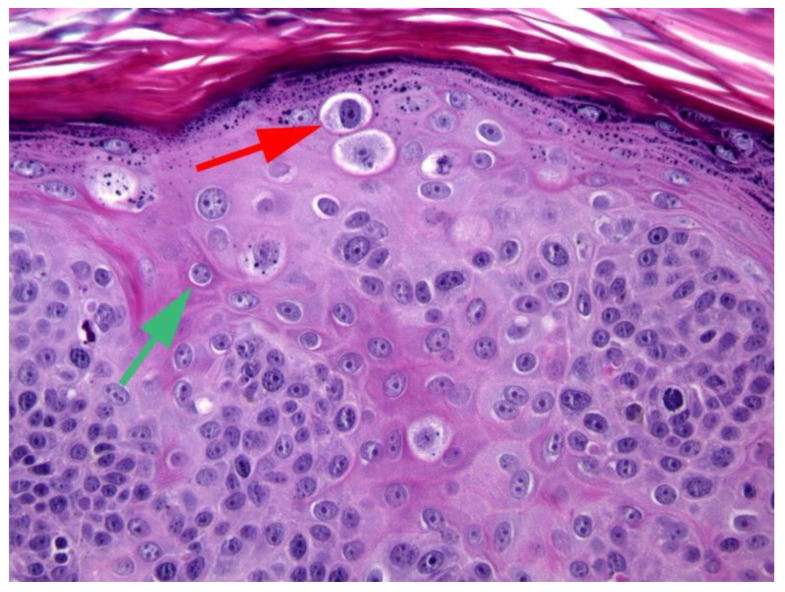
Canine cutaneous inverted papilloma. Canis familiaris papillomavirus (CPV) type 1 DNA was amplified from this lesion. Unlike papillomas caused by CPV6, papillomaviral changes can be subtle in warts caused by CPV1. In this case, enlarged cells with blue-grey cytoplasm (red arrow) and cells with perinuclear clearing (green arrow) are scattered throughout the lesion. Keratohyalin clumping is also visible and can be indicative of a papillomavirus etiology. HE 400×.

**Figure 6 pathogens-13-01057-f006:**
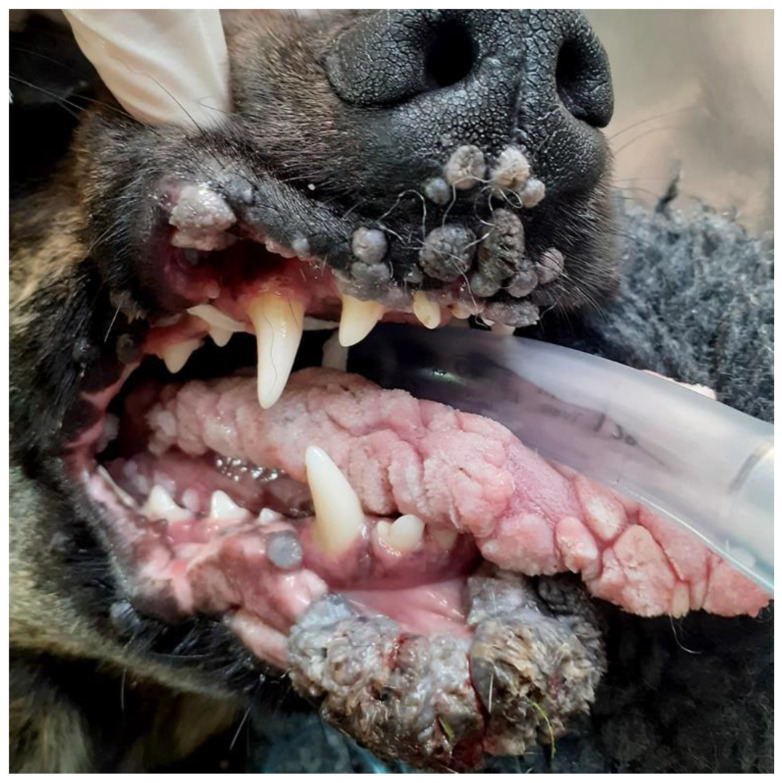
Persistent canine oral warts. The oral warts in this dog had been present for over 12 months. They are extensive, and new warts continued to develop throughout the 12-month period. Canis familiaris papillomavirus (CPV) type 1 and CPV26 DNA were amplified from the warts. Warts recurred after multiple debulking surgeries. The animal was eventually euthanatized due to the large warts involving an extensive area of the mouth. The failure to mount an immune reaction suggests a papillomavirus-specific immunodeficiency was likely in this case. Photo courtesy Dr. Susie Soulsby, Northland Veterinary Group, New Zealand.

**Figure 7 pathogens-13-01057-f007:**
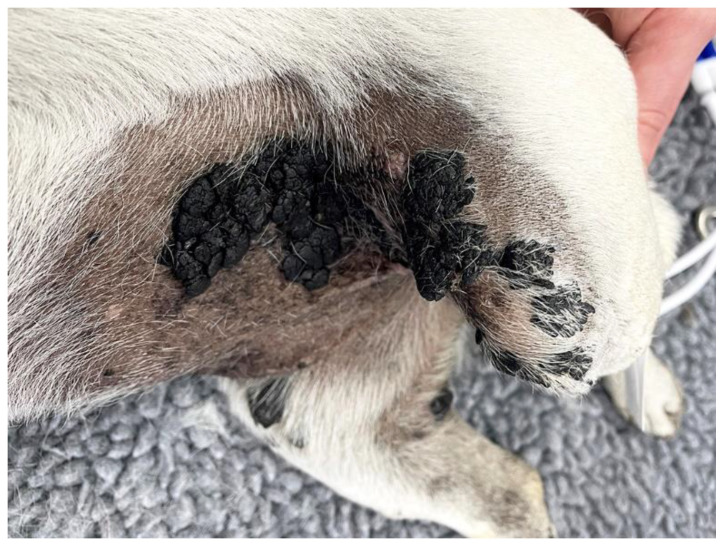
Canine viral plaques, pug. The plaques were mainly on the ventrum and medial aspects of the hind legs. Plaques are darkly pigmented. The plaques, in this case, are extensively larger and more exophytic than is typically seen for this condition. Canis familiaris papillomavirus type 24 DNA was amplified from this lesion. Photo courtesy Dr. Philippa Ravens, Small Animal Specialist Hospital, Australia.

**Figure 8 pathogens-13-01057-f008:**
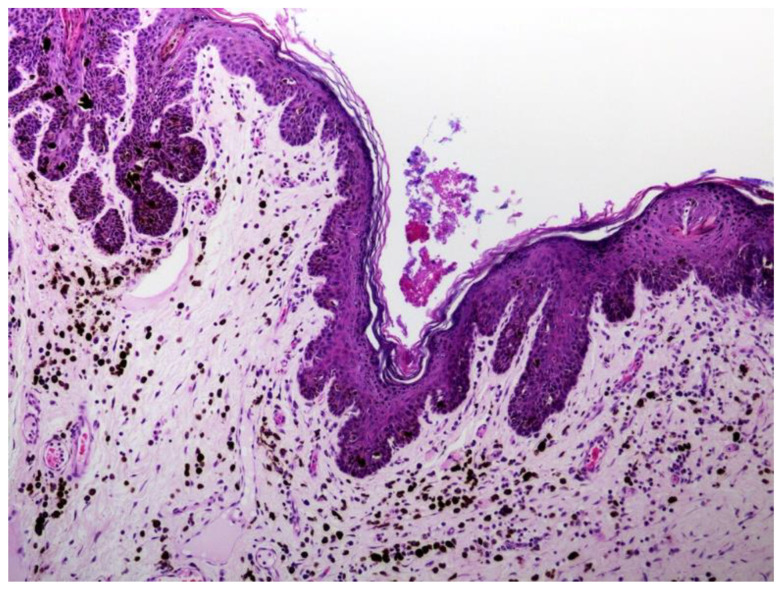
Canine viral plaque. The plaque comprises moderately thickened epidermis covered by mildly increased keratin. The epidermis has an irregular ‘scalloped’ appearance with large quantities of intraepidermal and dermal melanin pigment. Canis familiaris papillomavirus type 4 DNA was amplified from this lesion. HE 100×.

**Figure 9 pathogens-13-01057-f009:**
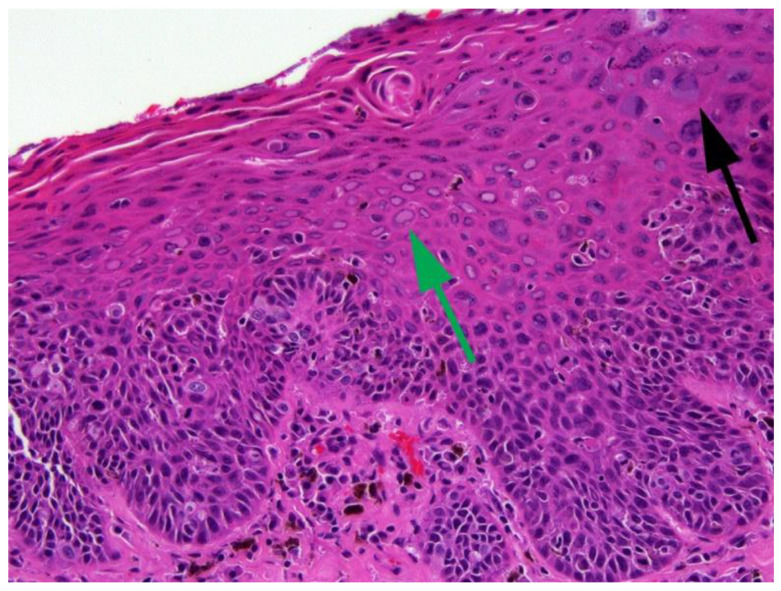
Canine oral squamous cell carcinoma. This dog presented with multiple oral squamous cell carcinomas that eventually necessitated the euthanasia of the animal. Histology reveals a well-differentiated squamous cell carcinoma containing enlarged cells with smudged blue cytoplasm (black arrow) as well as cells with prominent eosinophilic intranuclear inclusions (green arrow). Canis familiaris papillomavirus type 17 DNA was amplified from multiple lesions, and the neoplasms showed intense intranuclear and intracytoplasmic p16^CDKN2A^ protein immunostaining. HE 200×.

**Figure 10 pathogens-13-01057-f010:**
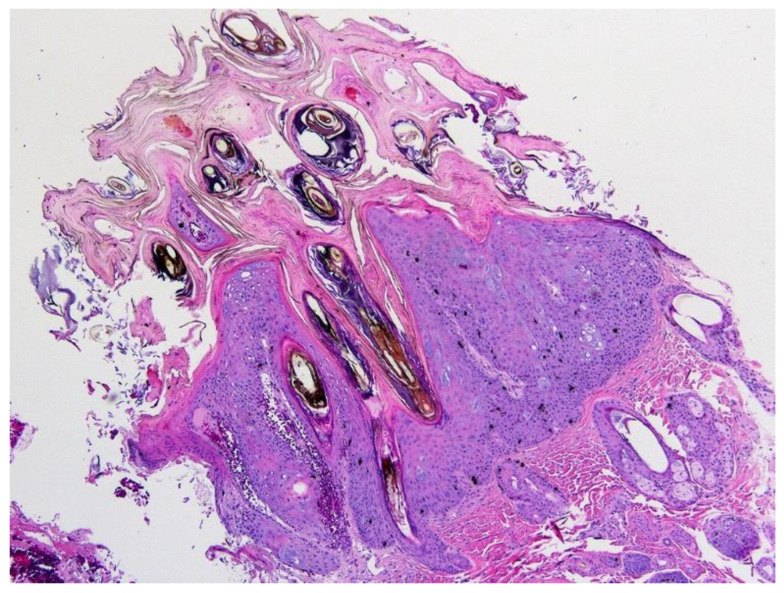
Feline cutaneous papilloma. This papilloma was visible as a small exophytic nodule overlying the bridge of the nose. Surgical excision was curative. The mass comprises markedly thickened epidermis thrown into folds. Papillomavirus-induced cell changes are prominent within the papilloma. Short sections of DNA from a novel Felis catus papillomavirus type were amplified from the lesion. HE 50×.

**Figure 11 pathogens-13-01057-f011:**
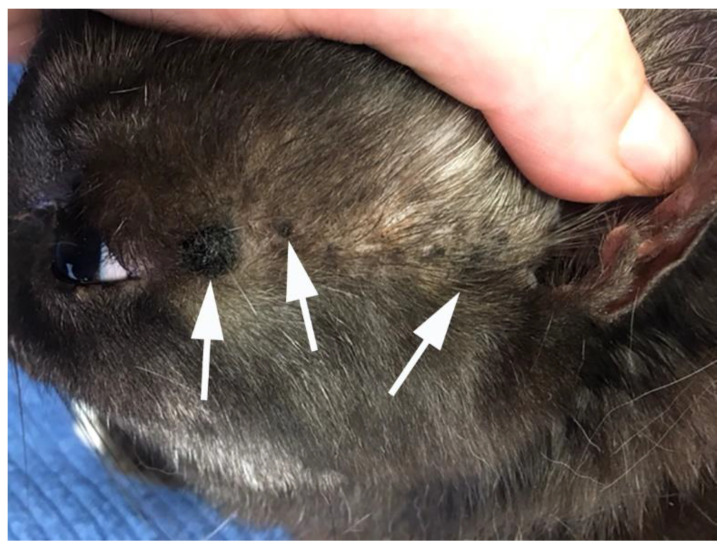
Feline viral plaques. Multiple dark, slightly raised plaques are visible on the head of this cat (arrows).

**Figure 12 pathogens-13-01057-f012:**
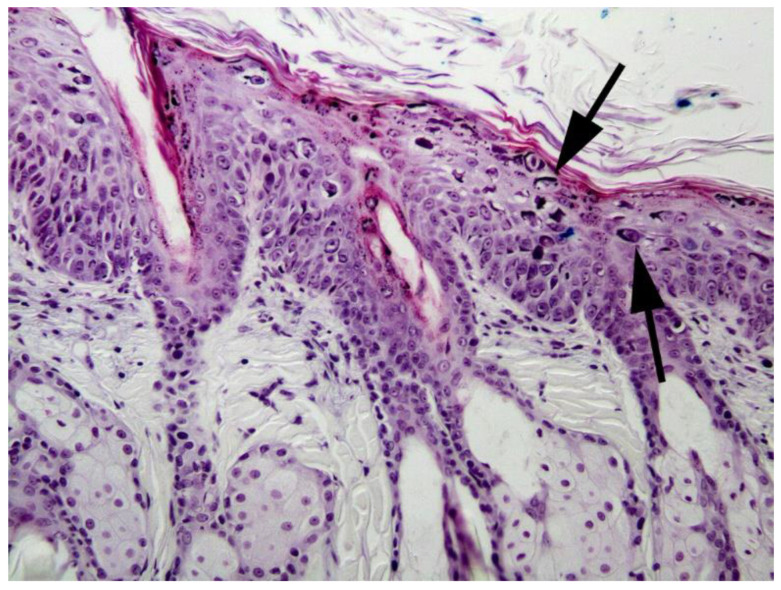
Feline viral plaque. This early plaque contains only moderate epidermal thickening and mild dysplasia. Numerous cells contain evidence of papillomavirus infection, including cells that have prominent brick-like, darkly basophilic intracytoplasmic inclusions (arrows). Felis catus papillomavirus type 3 DNA was amplified from this lesion. HE 100×.

**Figure 13 pathogens-13-01057-f013:**
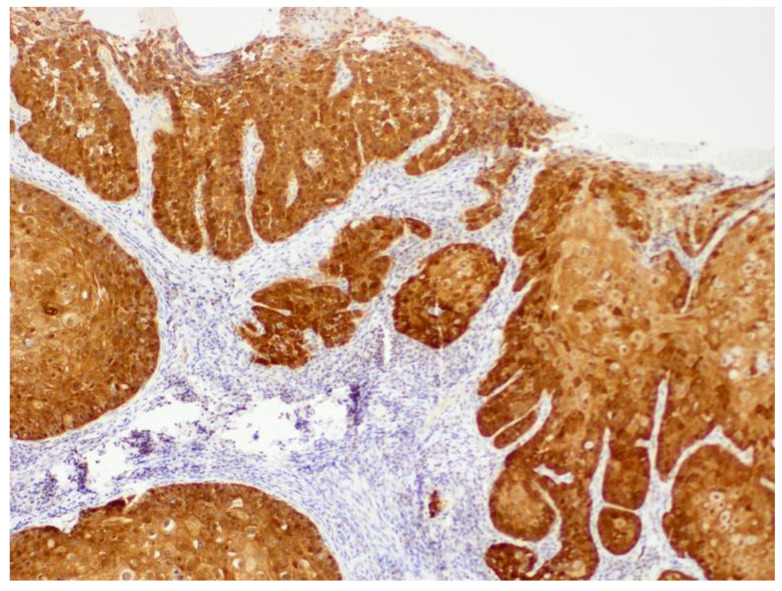
Feline cutaneous squamous cell carcinoma. Intense intranuclear and intracytoplasmic p16^CDKN2A^ protein immunostaining is visible within the neoplastic cells, including those in invasive trabeculae of cells. Felis catus papillomavirus type 2 DNA was amplified from this lesion. 200×.

**Figure 14 pathogens-13-01057-f014:**
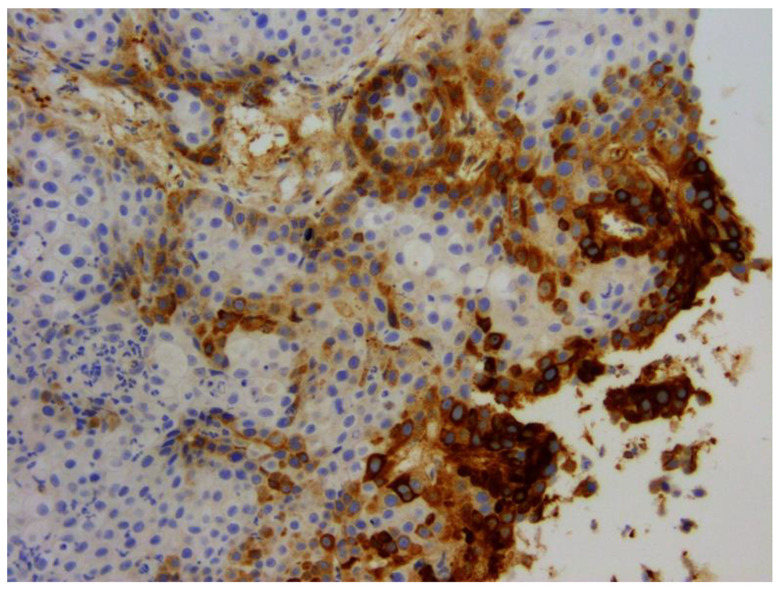
Feline oral squamous cell carcinoma. While some cells show intracytoplasmic p16^CDKN2A^ protein immunostaining (p16), most do not, and none of the cells show intranuclear immunostaining. This lesion did not contain amplifiable papillomavirus DNA. While variability in p16 can be visible within feline oral squamous cell carcinomas, the intense intranuclear and intracytoplasmic immunostaining that characterizes a PV-induced cutaneous lesion is rarely present in these oral cancers. 200×.

**Figure 15 pathogens-13-01057-f015:**
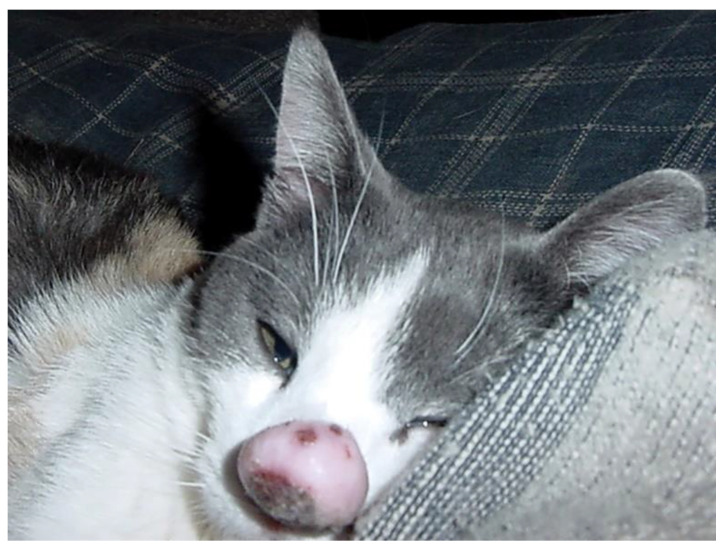
Feline sarcoid. The sarcoid appears as a fleshy mass protruding from the nasal philtrum. The sarcoid recurred following surgical excision and necessitated the euthanasia of the cat. Bos taurus papillomavirus type 14 DNA was amplified from the sarcoid.

**Table 1 pathogens-13-01057-t001:** Summary of the 4 canine and 3 feline papillomavirus genera and their associated lesions. Note that when the role of the papillomaviruses in lesion development remains uncertain, the lesion is not included in this table. CPV is Canis familiaris papillomavirus, FcaPV is Felis catus papillomavirus, SCC is squamous cell carcinoma, BCC is basal cell carcinoma.

Species	Papillomavirus Genus	Example Types	Associated Lesion
Dog	*Lambdapapillomavirus*	CPV1, 6	Oral and cutaneous viral papilloma [39,40]
	*Taupapillomavirus*	CPV2, 7	Cutaneous viral papilloma [40,41]
		CPV17	Oral SCC [42]
	*Chipapillomavirus*	CPV3, 4, 16	Cutaneous viral plaques [43] Cutaneous SCC [44]
	*Omegapapillomavirus*	CPV26	Co-infection in oral viral papilloma [45]
Cat	*Lambdapapillomavirus*	FcaPV1	Oral and cutaneous viral papilloma [46,47]
	*Dyothetapapillomavirus*	FcaPV2	Cutaneous viral plaques [48] Cutaneous SCC [49]
	*Taupapillomavirus*	FcaPV3, 4, 5	Cutaneous viral plaques [50,51] Cutaneous SCC [52] Cutaneous BCC [50]

## Data Availability

All studies referenced in this manuscript are contained within publicly searchable databases.

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
