# Peer review of "Papillomaviruses and Papillomaviral Disease in Dogs and Cats: A Comprehensive Review"

_pathogens, 2024, doi:10.3390/pathogens13121057_

Round 1
Reviewer 1 Report
Comments and Suggestions for Authors
This is a well written and comprehensive review.
It might be worth mentioning in the introduction that the viral genome is not always integrated in HPV-driven cancers (for example in skin and HN SCC) and this may be in alignment with what is seen in dog and cat neoplasms.
Otherwise, the topic of dog and cat PVs is covered and explained well, and suitably referenced.
Author Response
This is a well written and comprehensive review.
Thank you very much for the positive comments, the authors greatly appreciate this feedback.
It might be worth mentioning in the introduction that the viral genome is not always integrated in HPV-driven cancers (for example in skin and HN SCC) and this may be in alignment with what is seen in dog and cat neoplasms.
This is an excellent idea and this has been added as suggested and referenced (lines 74 and 95).
Otherwise, the topic of dog and cat PVs is covered and explained well, and suitably referenced.
Reviewer 2 Report
Comments and Suggestions for Authors
In this review, Munday and Knight bring new insights into Papillomaviruses, namely canine and feline types. In this regard, the authors describe the life cycle of PVs, together with the etiopathogenetic mechanism underlying the neoplastic and non-neoplastic diseases occurrence. Further, the immune response to PVs and the classification of these viruses are reviewed, along with the Papillomavirus-associated diseases of dogs and cats, including the clinical presentation, causative PV type, histology, and possible treatment options. A discussion on the potential use of vaccines to prevent PV-induced diseases concludes the review.
Overall, the writing is very good and well documented, using references up to date.
The quality of the English is commendable and I do not have much concern about the manuscript.
Minor concerns:
Throughout the manuscript, the authors use the references in the brackets at the beginning of the sentences, which for me is unclear. The reference is used for the previous sentence or for the sentence including the reference at the beginning? Please clarify this aspect.
Additionally, please uniformize the references throughout the manuscript. Eg: line 46 the reference is [2,3], while one row under (line 47) is in bold [4].
Line 44. The information given here about the genome structure is correct, but there is an exception, some PV possesses L3 gene, and this information should be introduced here. See 1) A Eriksson, H Ahola, U Pettersson, J Moreno-Lopez, 1988, Genome of the European elk papillomavirus (EEPV. Virus Genes, 1988 Mar;1(2):123-33, doi: 10.1007/BF00555932. 2) https://www.uniprot.org/uniprotkb/A0A1B2K237/entry
Line 118. Fig 1. The author should add the magnification. The image is original? If yes, state this. Additionally, use the same font (there are 2 different fonts).
Table 1 Line 169: The authors should add references where the association of a various CPV or Fca PV with a lesions is reported.
Figure 4, line 233. Add the staining method used and the magnification
Same with figure 8 and 9.
Line 360. From my point of view, the image used is not relevant for describing the p16 immunopositivity. Could the authors indicate where these intense intranuclear positivity is located? In the dermal layer, some fibroblasts display nuclear immunoreactivity.
Figure 10, line 376. Add the staining method used and the magnification
Figure 13. I cannot see any specific immunoreactivity in this picture. All cells are stained, I believe the primary antibody was too concentrated. I suggest to use a more specific image for p16 antigen immunodetection.
Supplementary, I suggest to introduce some phrases including the therapy with some Tyrosine kinase inhibitors, such imatinib, since clinical trial evaluating imatinib mesylate are available.
Author Response
n this review, Munday and Knight bring new insights into Papillomaviruses, namely canine and feline types. In this regard, the authors describe the life cycle of PVs, together with the etiopathogenetic mechanism underlying the neoplastic and non-neoplastic diseases occurrence. Further, the immune response to PVs and the classification of these viruses are reviewed, along with the Papillomavirus-associated diseases of dogs and cats, including the clinical presentation, causative PV type, histology, and possible treatment options. A discussion on the potential use of vaccines to prevent PV-induced diseases concludes the review.
Overall, the writing is very good and well documented, using references up to date.
The quality of the English is commendable and I do not have much concern about the manuscript.
Thank you very much for the positive comments – they are much appreciated by the authors.
Minor concerns:
Throughout the manuscript, the authors use the references in the brackets at the beginning of the sentences, which for me is unclear. The reference is used for the previous sentence or for the sentence including the reference at the beginning? Please clarify this aspect.
Additionally, please uniformize the references throughout the manuscript. Eg: line 46 the reference is [2,3], while one row under (line 47) is in bold [4].
The authors apologize for this error – the reference formatting has been corrected throughout.
Line 44. The information given here about the genome structure is correct, but there is an exception, some PV possesses L3 gene, and this information should be introduced here. See 1) A Eriksson, H Ahola, U Pettersson, J Moreno-Lopez, 1988, Genome of the European elk papillomavirus (EEPV. Virus Genes, 1988 Mar;1(2):123-33, doi: 10.1007/BF00555932. 2) https://www.uniprot.org/uniprotkb/A0A1B2K237/entry
This oversight has been corrected and referenced as suggested (line 55 new ref 2).
Line 118. Fig 1. The author should add the magnification. The image is original? If yes, state this. Additionally, use the same font (there are 2 different fonts).
The magnification has been added as requested (line 127). All figures are original – any reused ones would be indicated in the figure legend.
Table 1 Line 169: The authors should add references where the association of a various CPV or Fca PV with a lesions is reported.
This has been added within the table as suggested (line 183).
Figure 4, line 233. Add the staining method used and the magnification
Added (line 267)
Same with figure 8 and 9.
Added (line 363 and 396)
Line 360. From my point of view, the image used is not relevant for describing the p16 immunopositivity. Could the authors indicate where these intense intranuclear positivity is located? In the dermal layer, some fibroblasts display nuclear immunoreactivity.
Figure 10, line 376. Add the staining method used and the magnification
Added (line 418).
Figure 13. I cannot see any specific immunoreactivity in this picture. All cells are stained, I believe the primary antibody was too concentrated. I suggest to use a more specific image for p16 antigen immunodetection.
This photomicrograph has been swapped for another with better areas on non-staining cells.
Supplementary, I suggest to introduce some phrases including the therapy with some Tyrosine kinase inhibitors, such imatinib, since clinical trial evaluating imatinib mesylate are available.
Thank you for this idea. While the authors agree there is some evidence these may be useful for treating some SCCs, the authors are both pathologists rather than oncologists and fear that suggesting the most appropriate way to treat SCCs in dogs and cats isn’t really their expertise and such decisions probably are best left up to the oncologists who are more experts in this field.
Reviewer 3 Report
Comments and Suggestions for Authors
The manuscript is a review of papillomavirus (PV)-related diseases in dogs and cats. The content is well-structured, encompassing papilloma life cycle and carcinogenesis, immune response, classification, consequences, macroscopic pathology in cats and dogs. The review contains pictures of macroscopic lesions and histopathology pictures. The integration of canine and feline perspectives highlights the similarities and differences in PV pathogenesis, adding depth to the comparative review.
Although the review is complete. Some sections lack detailed citations, especially when making broad statements about disease prevalence and outcomes. Additionally, would be good to have a diagnostic section within the review. Are there any molecular diagnostic tools that could be used to detect these viruses? Perhaps the authors could add a bit of information regarding tools that could be used to do a good differential diagnosis.
Specific comments:
Lines 15-25: While the abstract is informative, consider highlighting the clinical significance of PV-related diseases to better engage the audience. Perhaps adding a sentence on the potential implications for veterinary practice and public health.
Lines 31-41: The introduction effectively sets the stage but could benefit from a brief mention of the zoonotic implications of PVs, even if limited.
Line 65: "However it should" → "However, it should"
Lines 43-72: Excellent overview of the life cycle, the explanation of E6 and E7 proteins is well detailed. Consider condensing this explanation to a table or figure. Authors may consider adding a scheme or diagram to visualize the integration and expression of PV oncogenes.
Lines 126-140: This section could be improved by discussing how immune evasion strategies differ between canine and feline PVs.
Lines 132-137: The variability in immune responses is intriguing but underexplored. Provide examples of factors influencing this variability.
Lines 141-154: The phylogenetic tree (Figure 2) is a strength, but the explanation of the classification system is repetitive and could be streamlined. Consider focusing on clinically relevant genera and their associated diseases.
Line 250: The discussion of azithromycin lacks recent evidence; consider citing newer studies or clarifying its controversial efficacy. The current reference is from 2008.
Lines 381-416: Expand on genetic predispositions in certain breeds and discuss potential implications for breeding practices. Poodles?
Lines 514-541: While the section is detailed, it could include a discussion of barriers to vaccine development, such as cost and market demand, to provide a holistic view. Would be worthwhile mentioning that methodologies used to develop vaccines against human PV could serve as good platforms as a starting point for canine or feline vaccines.
Author Response
The manuscript is a review of papillomavirus (PV)-related diseases in dogs and cats. The content is well-structured, encompassing papilloma life cycle and carcinogenesis, immune response, classification, consequences, macroscopic pathology in cats and dogs. The review contains pictures of macroscopic lesions and histopathology pictures. The integration of canine and feline perspectives highlights the similarities and differences in PV pathogenesis, adding depth to the comparative review.
Thank you for the positive reviews, the authors really appreciate the positive feedback.
Although the review is complete. Some sections lack detailed citations, especially when making broad statements about disease prevalence and outcomes. Additionally, would be good to have a diagnostic section within the review. Are there any molecular diagnostic tools that could be used to detect these viruses? Perhaps the authors could add a bit of information regarding tools that could be used to do a good differential diagnosis.
Including information on diagnosis is an excellent idea and this has been added (lines 226-241).
Specific comments:
Lines 15-25: While the abstract is informative, consider highlighting the clinical significance of PV-related diseases to better engage the audience. Perhaps adding a sentence on the potential implications for veterinary practice and public health.
This is an excellent idea. and this has been added (line 13-15).
Lines 31-41: The introduction effectively sets the stage but could benefit from a brief mention of the zoonotic implications of PVs, even if limited.
None of the canine or feline PVs are zoonotic and so this doesn’t seem to fit well in the abstract, but this information has been added later (line 197-199)
Line 65: "However it should" → "However, it should"
Changed as suggested (line 79)
Lines 43-72: Excellent overview of the life cycle, the explanation of E6 and E7 proteins is well detailed. Consider condensing this explanation to a table or figure. Authors may consider adding a scheme or diagram to visualize the integration and expression of PV oncogenes.
Thank you for the positive feedback. The authors considered this but feel a schematic may not be necessary and that the text probably is clear enough to explain the mechanisms.
Lines 126-140: This section could be improved by discussing how immune evasion strategies differ between canine and feline PVs.
Unfortunately to the authors’ knowledge, detailed information on the immune responses of either dogs or cats is not available.
Lines 132-137: The variability in immune responses is intriguing but underexplored. Provide examples of factors influencing this variability.
As with the previous comment, nothing is really known about the things that influence the variable time taken before an immune response is known. To the authors knowledge this is true in humans as well – some people rapidly resolve their warts while warts persist for years in others. Both responses are considered ‘normal’ and no studies have been done. Specific things that may reduce immunity and predispose to plaques are mentioned in these sections (lines 359-362 and 451-456)
Lines 141-154: The phylogenetic tree (Figure 2) is a strength, but the explanation of the classification system is repetitive and could be streamlined. Consider focusing on clinically relevant genera and their associated diseases.
This has been shortened as suggested with the genera just included in table 1 along with their associated diseases.
Line 250: The discussion of azithromycin lacks recent evidence; consider citing newer studies or clarifying its controversial efficacy. The current reference is from 2008.
I think this is the point of the discussion – the only evidence of efficacy is from 2008. No newer study has shown efficacy. Two references from 2021 and 2024 were included to show azithromycin doesn’t work, but obviously this was unclear so the sentence has been modified for clarity and references added (line 297-304).
Lines 381-416: Expand on genetic predispositions in certain breeds and discuss potential implications for breeding practices. Poodles?
Potential implications for breeding have been added (line 358-360). The authors are no aware of any breed predisposition for viral plaques in poodles.
Lines 514-541: While the section is detailed, it could include a discussion of barriers to vaccine development, such as cost and market demand, to provide a holistic view. Would be worthwhile mentioning that methodologies used to develop vaccines against human PV could serve as good platforms as a starting point for canine or feline vaccines.
Additional information has been added as suggested (lines 580-583).